# Dietary Habits and Gut Microbiota in Healthy Adults: Focusing on the Right Diet. A Systematic Review

**DOI:** 10.3390/ijms22136728

**Published:** 2021-06-23

**Authors:** Giulia Gibiino, Martina De Siena, Monica Sbrancia, Cecilia Binda, Vittorio Sambri, Antonio Gasbarrini, Carlo Fabbri

**Affiliations:** 1Gastroenterology and Digestive Endoscopy Unit, Ospedale Morgagni-Pierantoni, AUSL Romagna, 47121 Forlì, Italy; monica.sbrancia@auslromagna.it (M.S.); cecilia.binda@gmail.com (C.B.); carlo.fabbri@auslromagna.it (C.F.); 2Gastroenterology and Digestive Endoscopy Unit, Ospedale M.Bufalini, AUSL Romagna, 47521 Cesena, Italy; 3UOC di Medicina Interna e Gastroenterologia, Fondazione Policlinico Universitario A. Gemelli IRCCS, Università Cattolica del Sacro Cuore, 00168 Rome, Italy; martinadesiena@gmail.com (M.D.S.); antonio.gasbarrini@unicatt.it (A.G.); 4Unit of Microbiology, The Great Romagna Hub Laboratory, 47522 Pievesestina, Italy; vittorio.sambri@auslromagna.it; 5Unit of Microbiology, DIMES, University of Bologna, 40125 Bologna, Italy

**Keywords:** Mediterranean diet, microbiome, plant-based diets, short-chain fatty acids (SCFAs), Bacteroidetes, Firmicutes

## Abstract

Diet is the first to affect our intestinal microbiota and therefore the state of eubiosis. Several studies are highlighting the potential benefits of taking certain nutritional supplements, but a dietary regime that can ensure the health of the intestinal microbiota, and the many pathways it governs, is not yet clearly defined. We performed a systematic review of the main studies concerning the impact of an omnivorous diet on the composition of the microbiota and the production of short-chain fatty acids (SCFAs). Some genera and phyla of interest emerged significantly and about half of the studies evaluated consider them to have an equally significant impact on the production of SCFAs, to be a source of nutrition for our colon cells, and many other processes. Although numerous randomized trials are still needed, the Mediterranean diet could play a valuable role in ensuring our health through direct interaction with our microbiota.

## 1. Introduction

The human microbiota is the term used to describe the totality of bacteria, archaea, fungi, viruses, and protozoa that inhabits our organism [1]. Most of them (about 80% of the total) reside in the gastrointestinal tract in a proportion equal to about 1011–1012 microbes per millilitre (mL). Microbiota is composed prevalently by two phyla named Firmicutes and Bacteroidetes which together represent about 70–80% of the microbial totality; other phyla are Proteobacteria, Verrucomicrobia, Actinobacteria, and Fusobacteria. The microbiota establishes a mutualistic relationship with the host, and it can produce millions of active metabolites that will interact with our body’s complex networks. The intestinal microbiota should be conceived as a dynamic “organ” able to influence the absorption, metabolism, and storage of ingested nutrients [2]. It also exerts competitive phenomena with pathogenic microbes for the search of nutrients and of ecological niches colonization (barrier functions) [3] and modulates the functionality of the gastrointestinal tract, interacting with visceral sensitivity, motility, digestion, and substances secretion [4].

The microbiota can be considered as a fingerprint [5] unique for every human being with remarkable inter-individual and intra-individual variability depending on the surrounding environment such as geographical area, diet, and use of antibiotics. Microbiota influencers are indeed defined as external factors capable of altering the quantitative and/or qualitative composition of the microbiota [6] with a consequent body homeostasis alteration that is predisposed to diseases. Several recent studies are showing how single nutrients and also dietary regimens can have important consequences on gut microbiota composition. Mediterranean diet (MD) is linked to microbiota diversity and stability [7]. Physical activity is another factor influencing microbiota composition and future studies are needed to define the specific balance between diet and exercise for maintaining a healthy microbiota [8,9,10].

Dietary variations stimulated intestinal microbiota adaptations. During evolution, human dietary regimes have changed in relation to food and resources availability, environment pressures, and historical period. Modern society has had to face profound socio-economic and cultural changes in recent centuries since urbanization has drastically altered our lifestyle and habits, included food consumption. The increase in productivity consequently has led to an increase in the number of hours spent at work per day with drastic reduction in terms of times to dedicate to meal preparation. The “Western Diet”, characteristic of western countries is a diet rich in fat and low in fiber with higher consumption of processed and handled food. These meals are extremely rich in food additives such as emulsifiers and artificial sweeteners that alter the intestinal microbiota, predisposing them to various chronic and pre-cancerous pathologies such as cardiovascular and metabolic disease. Western diet seems to determine an increase of proinflammatory bacterial genes expression [11] such as metalloproteases (MMP-2) and nitric oxide synthetase (iNOS). Numerous studies conducted on animal models showed how emulsifiers (carboxymethylcellulose and polysorbate) administration can cause microbic alterations [12] with important microbiota changes (increase in Proteobacteria and *Escherichia coli* and reduction of *Bacteroides* and Clostridia). Furthermore, artificial sweeteners (sucralose, aspartame, and saccharin) would determine, in predisposed individuals, insulin resistance through an increase in *Bacteroides*, Clostridia, Enterobacteriaceae levels [13,14]. 

Clearly, however, not all populations in the “westernized” part of the world are exposed to this diet; on the contrary, the countries affected by the Mediterranean Sea are affected by it in terms of foodstuffs; in addition, a certain part of the world’s population is adopting vegetarian or vegan diets.

Mediterranean and vegan diet, rich in protective foods and beneficial substances, exert an anti-inflammatory effect [15] on our organism. 

Several studies have stated the impact of certain nutrients administered as supplements such as inulin [16] or omega-3 polyunsaturated fatty acids (PUFAs) [17], but these are not usually consumed in isolation, and we need clear reference points to assemble dietary and lifestyle habits that can be applied in clinical practice [18]. Despite numerous publications on the topic of diet and microbiota, there is still no clear interpretation and recommendation as to which dietary regimen should be adopted by the adult population not affected by specific diseases to preserve the state of eubiosis.

The purpose of this systematic review was to undertake an update on randomized clinical trials (RCTs) and cross-sectional studies including healthy adults and reporting the effect of an omnivorous and varied diet on gut microbiota composition. We assessed the difference in gut microbiota composition, from phylum to species reported; furthermore, we adopted the SCFAs’ production as a secondary outcome as direct representation of metabolic change in the gut function.

## 2. Results

### 2.1. Study Characteristics 

Included studies were 13 cross-sectional studies and three randomized trials. Studies eligible for the review are reported in Table 1. 

The study population was equally represented by healthy males and females, with age ranging from 18 to 75 years. 

A single study conducted by Stefani S [19] was conducted only on female subjects, involving 240 healthy women in Indonesia. 

For each study, we considered BMI as a nutritional measure that was mostly reported but not always considered as a diagnostic for overweight or obesity and metabolic syndrome. 

Geographical setting regarded Italy for seven studies [19,20,21,22,23,24] and out of them, two studies [20,21] started from the same study population to evaluate different and complementary outcomes; a European study evaluated populations in Poland, Netherlands, France, and UK [25]; other European countries mentioned are Greece [26], Germany [27], and Spain [28,29]. Brazil [30], USA [31], Indonesia [19], and China [32,33] were represented as well. 

The primary intervention considered in all studies was an omnivorous dietary regimen, lasting at least 3 months, sometimes proposed as a single-arm intervention [24,28,29,34] and sometimes in parallel with a vegan or vegetarian diet. When specifically expressed by the authors, we marked that the diet was classified as “Mediterranean type” [20,21,24,25,26,29,35,36,37]. In one study, a varied diet with three isocaloric formulations with different fat contents was considered [33]. The adherence to the assigned regimen was measured differently, ranging from validated questionnaires or scores up to cases of food diaries reported by patients. 

### 2.2. Microbiota Composition in Omnivore Population

A significant modification of the gut microbiota after exposure to omnivore diet was showed in 11 [20,21,22,25,26,28,29,30,32,33,35,36,37,38] out of 16 studies. In all the studies considered, an analysis of the microbiota was carried out by molecular sequencing techniques, except for a single study using an analysis based on bacterial count in faecal samples [22]. Exposure to an omnivore diet resulted in significant alteration in the composition of the microbiota at different taxonomic levels. Genera significantly influenced by the exposure to the diet interventional were *Ruminococcus, Streptococcus, Staphylococcus*, all part of the Firmicutes phylum. Other genera mentioned were *Prevotella, Enterorhabdus, Lachnoclostridium, Parabacteroides,* and *Bacteroides*, which are instead part of the Bacteroidetes phylum. Some genera belonging to Firmicutes and Bacteroidetes also showed interesting results in the comparison between diets with different fat content [33]. In addition, Federici et al. [22] showed significant values of *Corynebacteria* (ph. actinobacteria) while Franco-de-Moraes et al. [30] illustrated *Succinivibrio* and *Halomonas* (phylum: Proteobacteria).

Significant results at species level were identified as *B*. *fragilis* [20], *Clostridium cluster XVIa*, and *Faecalibacterium prausnitzii* [29]; Ghosh et al. also described specific “Diet Positive Operational Taxonomic Units (OTUs)”, responding to increasing adherence to Mediterranean diet and were represented by *Faecalibacterium prausnitzii, Roseburia hominis, Bacteroides thetaiotaomicron, Prevotella copri,* and *Anaerostipes hadrus* [25].

### 2.3. Short-Chain Fatty Acids Production

Six studies evaluated the production of short-chain fatty acids (SCFAs), with significant results in almost all except for the study carried out by Wu et al. [31]. De Filippis [21] found a faecal SCFA profile increasing with Mediterranean diet and plant-based diets with a specific positive correlation with *Prevotella* genus. The impact of the Mediterranean diet on these metabolic products is similarly reported by Gutierrez-diaz et al. [28] and Mitsou et al. [35].

The clinical trial conducted by Pagliai et al. [36] showed not overall correlation between MD and SCFAs, but more interesting they showed a mean variation of each SCFA (post–pre diet) with an opposite and statistically significant trend for propionic acid (*p* = 0.034) with an increase of 10% after Mediterranean diet consumption. 

## 3. Discussion

This systematic review is an update about the fine interplay between diet and microbiota, to show evidence and key points for the best omnivorous diet in healthy adults. The composition of the gut microbiota of the healthy young adult is a point of great interest. In fact, it is well known that microbiota changes over time and reaches its maximum microbial biodiversity at that stage. A recent study showed that healthy young adults have a microbiota that can restore itself after broad-spectrum antibiotic treatment due to resilience [39]. In the elderly, this endocrine tissue becomes compositionally unstable and it is characterized by less diversity; these can be linked to age-related endocrine modifications and immune suppression [1].

A nutritional regimen today cannot ignore the processing of food and the presence of additives and contaminants. Current ultra-processed food includes high calorie density able to satisfy consumer taste and western habits. We are used to thinking about the best nutrients as best absorbed by our cells, but the discovery of the huge variety of commensals living in our gut could have a revolutionary impact on this. Indeed, the undigested material can be a substrate for the nutrition of microbiota, leading to its diversity. Food with a low bioavailability of proteins, carbohydrates, and lipids or phytochemicals could result in higher levels of nutrient delivery to the microbiota while the absorption by the host would be limited to few calories [40]. This interesting theory about the best food design able to feed out microbiota, purposed by Ercolini D et al. [41], could give an innovative concept of the best human diet. Furthermore, the food most present on western tables is enriched by chemical additives, usually considered safe but still not correctly evaluated in terms of microbiome interaction. In some cases, microbial transformation of dietary bioactives may have undesirable consequences [42,43]. Two recently identified examples are particularly noteworthy: artificial sweeteners and emulsifiers. Artificial sweeteners are used to provide flavour without calories, in order to reduce the obesity widespread in the world, but while interacting with our gut microbes, they could lead to unexpected glucose intolerance themselves. This was showed in animal models [44], but is still not well evaluated in a human host. Emulsifying agents are detergent-like compounds added to processed foods to keep particles in suspension, particularly during storage. The importance of this is that they affect our mucosal layer and bacterial translocation [45], which are well-known key players in bowel inflammation, metabolic syndrome, and carcinogenesis. Our systematic review assessed omnivorous diets including studies that did not specify further information about food processing in the current age. It was highlighted when it was specified that the omnivorous diet was Mediterranean, considering that this type of diet was linked to several beneficial effects on our microbiota [46]. The future personalized nutrition probably will be based on gut microbiota interaction, with consequences on agriculture and food production modalities, aware that the food should be the best one designed to feed our microbiota.

A second important point regarding evidence on gut microbiota and diet is about techniques for studying gut microbiota. We know that the introduction of molecular techniques has made it possible to define many microbes that were not known before because they could not be cultivated with standard techniques. Before the advent of next-generation sequencing (NGS) techniques, the only techniques allowed included cultivation of individual bacteria and studies of interactions by co-culture of microbial consortia. With the introduction of DNA sequencing technology, it became possible to have a precise and rapid taxonomic identification of individuals within those communities. Sequencing analysis of the 16S rRNA gene was originally performed by cloning the full gene into plasmid vectors, transforming it into suitable hosts (usually *Escherichia coli*), and sequencing it [47]. Then other methods like Southern blotting and in situ hybridization made use of the 16S rRNA clone libraries to identify members of complex microbial communities [48]. Today, one standard method for determination of gut microbiome composition is performed by isolation of total DNA from samples, PCR amplification of regions within universally conserved 16S/18S rRNA genes, followed by high-throughput sequencing of those amplicons. This technology has eliminated the need for cloning individual genes, blotting for specific RNA, or cultivating individual microbes to identify members of a community [49]. Beyond the technological variety available to the centers dealing with this, we still lack a specific validated reference library, capable of being the basis of comparison to define the healthy microbiome. Therefore, there is no standard microbiome ecology that all healthy people share. However, because of this high variability among individuals, extreme caution must be taken in interpreting results from fewer than hundreds of people, and the reference range approach that has worked for blood tests will not work for the microbiome [50]. For all this, we should admit our current strong limitation in carrying out research on the microbiota field and therefore on all the agents that interact with it, first the diet. In other words, we should address the technology according to a shared and validated rule, currently not available.

A third important point usually not considered when studying microbiota is its interaction with our innate and adaptive immune system. The MHC encodes for the alleles of HLA class I and class II loci, which are the most polymorphic genes in humans, and which determine the specificity of T lymphocyte and natural killer (NK) cell responses, including against the commensal bacteria present in the human gut. This system has been considered among the genetic factors that can determine our wealth of intestinal microbiota and there is an extreme variability inherent in each of us. As an evolving crosstalk, the microbiota regulates our immune system throughout life and vice versa, the HLA system also changes its composition. We do not know exactly the mechanisms that govern the interaction between HLA and intestinal commensals, potentially including immune-mediated elimination or directly affecting bacterial adhesion [51]. Recent studies began showed interesting links between dietary microbiota modulation and host immunity. Western-style diets adversely impact host immunity [52]. Some examples have been recently reported by Zheng et al. [53], since a diet high in saturated fats increases the levels of taurocholic acid, a secondary bile acid, and in turn fosters the expansion of *Bilophila wadsworthia*. This pathobiont promotes Th1 type immune responses and increases susceptibility to colitis in IL10–/– mice [54]. Moreover, dietary long-chain fatty acids may exacerbate autoimmunity in the central nervous system (CNS) by modulating the gut microbiome and metabolome [55]. Surely this factor will have to be clarified by future research and will lead to a more personalized diet, tailored by gene-expression and specific microbiota composition.

## 4. Materials and Methods

### 4.1. Search Strategy

The papers to be included were sought in the PubMed, Scopus, Clinicaltrials.gov, Web of Science, and Cochrane Library databases in September 2020. The search terms used alone or in combination were: “Diet” OR “diet intervention” OR “dietary habits” OR “Mediterranean diet” OR “Omnivore diet” OR “Vegetarian and vegan diet” OR “Western diet AND human gut microbiota assessment” OR “Microbiome difference” OR “Microbiota changes” OR “Metabolomic changes” OR “Bacterial intestinal composition” OR “faecal microbiota” OR “faecal metabolic profile” OR “short-chain fatty acid production” OR “diet-enterotypes”. 

We included all the clinical investigations involving the effect of a matching specific dietary regimen on gut microbiota assessment in healthy adults. We assessed RCTs comparing dietary regimens as randomized controlled or randomized cross-over designed; we also considered cross-sectional studies evaluating the influence of a specific diet on gut microbiota composition. 

### 4.2. Eligibility Criteria 

To be eligible for inclusion, studies had to include microbiota assessment in healthy adults following specific dietary regimens: omnivore or including several nutrients. The study population had to be negative to any specific disease diagnosis, such as obesity, diabetes, inflammatory bowel disease (IBD), irritable bowel syndrome (IBS), cancer, and reported gastrointestinal or systemic diseases of any kind. The studies had to assess microbial composition to define significant differences in phyla, genera, or species levels after almost 3 months of omnivore diet. Another outcome considered to include in studies was the evaluation of the direct metabolic modification measured by short-chain fatty acids (SCFAs) production. Only studies published in English over the previous 10 years were considered. Abstracts or conference communication were not included. The papers were selected by two independent reviewers (G.G. and M.D.S). We excluded studies conducted on animal populations or children. We did not consider studies on non-healthy adults, such as patients affected by obesity, cardiovascular disease, cancer, or gastrointestinal disease such as inflammatory bowel disease (IBD), irritable bowel syndrome (IBS), coeliac disease. Then, we excluded all the investigations regarding supplements or single nutrients (inulin supplementation, whole-grain assumption, gluten, Low FODMAPs, and omega3) to select only specific whole dietary regimens. Reviews or studies not reported as randomized controlled trials, or cross-over or cross-sectional studies, were not considered.

### 4.3. Study Selection and Data Extraction

The selection of articles for inclusion in the review was undertaken in two stages. The first stage involved screening the title and abstracts of the search results against the eligibility criteria. In the second stage, the full articles of papers selected in the title/abstract screening stage were screened to confirm that they met the eligibility criteria. At both stages, each article was screened independently by two authors (G.G. and M.D.S). Disagreement in eligibility status between the first two authors were resolved by a third author or mutual discussion. 

Papers were selected using the Preferred Reporting Items for Systematic Reviews and Meta-Analyses (PRISMA) flowchart and the PRISMA checklist [56] A Preferred Reporting Items for Systematic Reviews and Meta-Analysis (PRISMA) flow diagram is summarized in Figure 1.

Two authors independently extracted data from each study. Data extracted included details of study design, participants (sample size, country/region, BMI), details of dietary regimen, and the measure of diet adherence and outcomes.

### 4.4. Quality Assessment of Studies

The quality assessment was based on the Cochrane Collaboration tool [16] for randomized trials and the Newcastle–Ottawa scale [57] for cross-sectional studies. Reporting details were conducted by two authors (G.G. and M.D.S.). The mean score for cross-sectional studies was 5/10 but there was no comparison between respondents and non-respondents as previously reported by Trefflich et al. [27], who performed a similar systematic review on vegetarian or vegan diet. Tables reporting the study assessment for respective scales are available as Appendix A. 

### 4.5. Microbiota Data Reports

Of the reviewed studies, evidence on microorganisms reported at phylum, family, genus, and species level was considered. Microbiota of subjects following omnivore diet was the only diet adopted or compared with other dietary regimens, mainly represented by vegetarians or vegans, assessed as control group.

## 5. Conclusions

The diet we consume is our interface with the world and its effects on our intestinal microbiota has an impact on our state of health and disease. New evidence increasingly correlates the complicated relationship and connection between the composition of our microbiome and many tissues of various functions [58]. We summarized some studies showing a significant impact on some bacterial genus of a rich and varied diet, often framed as Mediterranean. However, the variability in microbiota study techniques and its interactions with dietary and genetic factors not yet well defined make the evidence available today too uncertain. Further randomized human trials and research are needed to define the gut microbiota targets of our diet and how to modulate them with the most suitable combination of nutrients.

In conclusion, Hippocrates, father of medicine, already 2500 years ago ruled: “let food be your medicine and medicine be your food” emphasizing the importance of proper nutrition in the health of the individual. As it is now evident, it is a task of our generation of physicians to clarify what is the best diet for the human microbiota considering that the “diet of the future” must be not only functional but also environmentally sustainable.

## Figures and Tables

**Figure 1 ijms-22-06728-f001:**
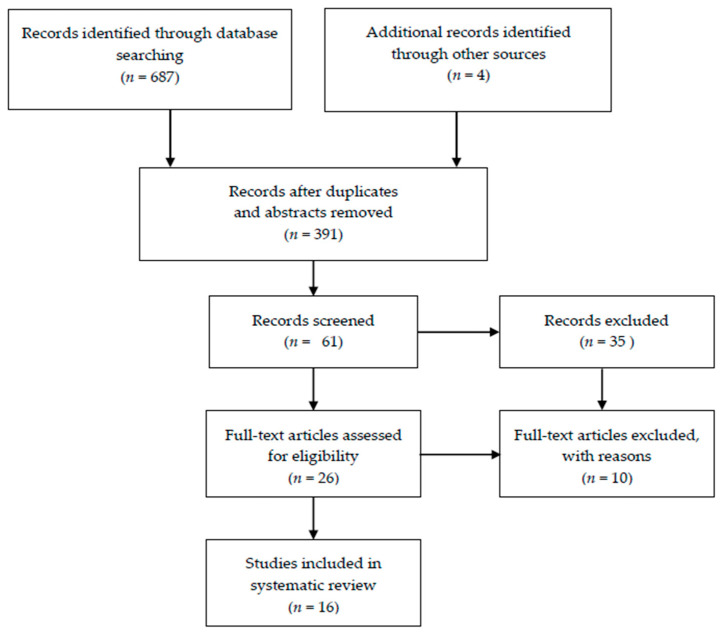
Preferred Reporting Items for Systematic Reviews and Meta-Analysis (PRISMA) flow diagram for the systematic review.

**Table 1 ijms-22-06728-t001:** Summary outline of the studies included in the systematic review.

Author, Year, Reference	Study Design	Study Population (Males, Mean or Median Age)	BMI (kg/m^2^) Mean or Median Value	Diet Intervention	Region	Dietary Assessment	Microbiota Assessment and Metabolomic Analysis	Outcome 1 Significant Difference in Microbiota Composition	Outcome 2 Short-Chain Fatty Acid (SCFAs) Production Variation
Ferrocino I, 2015 [20]	Cross- sectional	153, M 76, 18–55 ys	>18	51 vegetarians, 51 vegans and 51 omnivores ° >12 months *	Italy	Self-reported	Real-time quantitative PCR and rRNA-DGGE	Phylum: Bacteroidetes Genus: *Bacteroides* Species: *B. fragilis*	NR
De Filippis F, 2016 [21]	Cross- sectional	153 M 64, 18–55 ys	>18	51 vegetarians, 51 vegans and 51 omnivores ° >12 months ¹^,^*	Italy	Dietary score based on tertiles. The Healthy Food Diversity index (HFD)	16S rRNA sequencing + metabolomic analysis using gas-chromatography mass spectrometry-solid-phase microextraction	Phylum: Firmicutes Genus: *Ruminococcus* Genus: *Streptococcus*	Faecal SCFA profile increasing with MD and plant-based diets. (*Prevotella* was the only Bacteroidetes having positive correlations with SCFA
Wu GD, 2016 [31]	Cross- sectional	31, NR	NR	15 vegans and 16 omnivores for ≥6 months	USA	Food frequency questionnaire (FFQ)	16S rRNA sequencing +metabolomic analysis of SCFAs faces with nuclear magnetic resonance (NMR) spectra of faecal water	No significant difference at genus level	No significant effect on the levels of faecal SCFAs despite plant-based diet increase
Gùtierrez-Diàz I, 2016 [28]	Cross- sectional	31, M 8, 42.1 ± 10.9 ys	NR	31 subjects following Med Diet ° ≥ 6 months	Spain	Mediterranean Diet Score	16S rRNA sequencing + metabolomic analysis faceal SCFAs detected by gas chromatographymass spectrometry (MS)	-Phylum: Bacteroidetes, Genus: *Prevotella* -Lower levels of Phylum Firmicutes and Genus *Ruminococcus*	Higher concentration of faecal propionate and butyrate
Gùtierrez-Diàz I, 2017 [29]	Cross- sectional	74, M 20, 50 ≥ 65 ys	25–30	74 subjects following Mediterranean diet °	Spain	Med Diet Score	16S rRNA sequencing and UPLC-ESI-MS/MS method for phenolic metabolytes analysis	-Phylum: Firmicutes Genus: *Clostridium* Species: Cl. XVIa -Phylum: Firmicutes Genus: *Faecalibacterium* Species: *F. prausnitzii*	NR
Federici E, 2017 [22]	Cross- sectional	29, M 14, 39 ± 10 ys, 33 ± 7 ys and 41 ± 9 ys, respectively	20.7 ± 2.2, 22.3 ± 2.2 and 22.6 ± 1.7, respectively	12 vegetarians, 10 vegans and 7 omnivores for >12 months	Italy	7-day weighed food diary	Faecal microbial counts	-Phylum: Actinobacteria Genus: *Corynebacteria* -Phylum: Firmicutes Genus *Staphylococcus*	NR
Franco-de-Moraes AC, 2017 [30]	Cross- sectional	268, M 123, respectively 49.6 ± 8.5 ys, 49.6 ± 8.6 ys and 49.1 ± 8.2 ys	<40	66 strict vegetarians, 102 lacto-ovo-vegetarians, and 100 omnivores for the last 12 months	Brasil	NR	16s RNA gene analysis	-Phylum: Protecobacteria Genus: *Succinivibrio* - Phylum: Protecobacteria Genus: *Halomonas*	NR
Mitso E.K, 2017 [35]	Cross- sectional	100, M 48, 41.27 ± 13.33 ys	27.29 ± 4.48	100 following Mediterranean diet ° (3 tertiles of adherence: low tertile, medium, tertile and high tertile)	Greece	Food Frequency Questionnaire (FFQ) and MedDiet Score	16s rRNA sequencing and metabolomic analysis were performed with capillary gas chromatography for faecal SCFAs	Phylum: Bacteroidetes Genus: *Bacteroides* + Increase of *C. albicans*	Med Diet positively linked to total SCFA
Losasso C, 2018 [23]	Cross- sectional	101, M 33, 42.5 ± 13.0 ys	23.8 ± 4.4	Vegans 26, vegetarians 32, and omnivores 43 for >12 months	Italy	Food frequency questionnaire (FFQ) and 24 h dietary recall	16s rRNA sequencing	No difference for bacterial community composition	NR
Stefani, S 2018 [19]	Cross- sectional	240 healthy women, 38.0 (31.0–44.0) ys	24.9 ± 49.5	Two groups of 120 women of West Sumatera and West Java provinces following predominantly animal- or plant-based traditional diets	Indonesia	2-day-repeated 24-h food recalls	DNA extraction and quantification of *Bifidobacterium* DNA using the *Bifidobacterium* sp. standard primer and using Real-Time PCR System	No significant alteration of genus *Bifidobacterium*	NR
Pagliai G, 2019 [26,36]	Randomized cross-over	23, M 7, 58.6 ± 9.8 ys	31.06 ± 0.67 and 30.10 ± 0.61 for the two groups	23 omnivorous ° entrolled: 11 following low-calorie Mediterran Diet and 12 Vegan Diet for three months and then crossed	Italy	Dietary randomization	16s rRNA sequencing and gas chromatography–mass spectrometry system for SCFAs	Phylum: Bacteroidetes Genus: *Enterorhabdus*, *Lachnoclostridium*, and *Parabacteroides*	Mean variation of each SCFA Increase of 10% of propionic acid
Trefflich I, 2019 [27]	Cross- sectional	72, 36 M, 37.5 (32.5–44.0) and 38.5 (32.0–46.0) ys, respectively	22.9 (± 3.2) and 24.0 (±2.1)	36 vegan and 36 omnivorous participants following diet for >12 months	Germany	NR	16S rRNA (rRNA) gene sequencing	Modest differences Not significant between vegans and omnivores at phylum, family, genus, and species level.	NR
Wang F, 2019 [32]	Cross- sectional	36, 28.1 ys	NR	36 adults following a vegan (12), a lacto-ovo vegetarian (12), or an omnivorous diet (12) for > 6 months	China	Nutrition System of Traditional Chinese Medicine Combining with Western Medicine, version 11.0	16s rRNA gene analysis	Phylum: Bacteroidetes Genus: *Bacteroides*	NR
Luisi MLE, 2019 [24]	Cross- sectional	36, M 17, 41.4 ± 14.42 and 52.1 ± 13.04 years, respectively for cases and controls	Cases ≥25 Controls 18.5 and 24.9	36 following typical MD ° and cases receiving a low-calorie MD for 3 months; both cases and controls utilized 40 g/die of HQ-EVOO as the only cooking and dressing fat	Italy	NR	dsDNA extracted from all the samples	No significant modification measurable	NR
Wan Y, 2019 [33,38]	Observer- blinded, RCT	217, M 114, Respectively, 23.3 (3.4), 23.6 (4.0) ys and 23.4 (4.1) ys	21.7 (2.6)	Lower-fat diet (73), moderate-fat diet (73) and higher-fat diet (71) for 6 months	China	Daily diary	16s RRNA sequencing and mass spectrometry system for SCFAs	-Lower-fat diet associated with increased Phyla: Firmicutes Genus: Blautia and Faecalibacterium -Higher-fat associated with increased Genus: Alistipes and Bacteroides (phylum: Bacteroidetes).	Total SCFAs significantly decreased in the higher-fat diet group in comparison with the other groups.
Ghosh TS, 2020 [25,37]	Randomized, multicentre, single-blind, controlled trial	612, M 286, Median age 65-75 ys	Median value 26.8 (18.8- 44.6) and 26.9 (18.5–46)	289 controls and 323 in MedDiet ° for 12 months	UK, France, Netherlands, Italy, and Poland	Adherence scores to the MedDiet, based on the NU-AGE Food Based Dietary Guidelines (FBDG)	DNA and 16S rRNA gene sequencing	“Diet Positive OTUs”: Phylum: Firmicutes Genus: *Faecalibacterium* Species: *faecalibacterium* *prausnitzii* -Phylum: Firmicutes Genus: *Roseburia* Species: *Roseburia hominis* -Phylum: Firmicutes Genus: *Eubacterium* -Phylum: Bacteroidetes Genus: *Bacteroides* Species: *Bacteroides thetaiotaomicron* -Phylum: Bacteroidetes Genus: *Prevotella* Species: *Prevotella copri* -Phylum: Firmicutes Genus: *Anaerostipes* Species: *Anaerostipes hadrus*	NR

* Authors used data of the same study population. ¹ The majority of vegan and vegetarian subjects and 30% of omnivore subjects had a high adherence to the Mediterranean diet. ° Omnivore diet considered as a Mediterranean type (MD). BMI, Body Max Index. NR, Not reported. rRNA DGGE, Ribosomal RNA Denaturing Gradient Gel Electrophoresis.

## Data Availability

Not Applicable.

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
