# Peer review of "Dietary Habits and Gut Microbiota in Healthy Adults: Focusing on the Right Diet. A Systematic Review"

_ijms, 2021, doi:10.3390/ijms22136728_

Round 1

Reviewer 1 Report

 I would like to congratulate the authors for the made effort in their study. The present manuscript by  Gibiino et al., analyzed “Dietary habits and gut microbiota in healthy adults: focusing on the right diet. A systematic review”. The authors with this systematic review try to better understand the impact of an omnivorous diet on the composition of the microbiota and the production of short-chain fatty acids (SCFAs). However, the authors should better show the existing problems and discuss to light of these and the existent literature (e.g. Rinninella et al. Food Components and Dietary Habits: Keys for a Healthy Gut Microbiota Composition. Nutrients. 2019 Oct 7;11(10):2393. doi: 10.3390/nu11102393; Cronin et al. Dietary Fibre Modulates the Gut Microbiota. Nutrients. 2021 May 13;13(5):1655. doi: 10.3390/nu13051655). Moreover, the specific correlation between diet and other confunding factors, such as physical activity, supplements (Donati Zeppa et al. Mutual Interactions among Exercise, Sport Supplements and Microbiota. Nutrients. 2019 Dec 20;12(1):17. doi: 10.3390/nu12010017; Dorelli et al. Can Physical Activity Influence Human Gut Microbiota Composition Independently of Diet? A Systematic Review. Nutrients. 2021 May 31;13(6):1890. doi: 10.3390/nu13061890; Aya et al. Association between physical activity and changes in intestinal microbiota composition: A systematic review. PLoS One. 2021 Feb 25;16(2):e0247039. doi: 10.1371/journal.pone.0247039).
The authors have correctly applied methodology for this Sistematic review, however should be detailed on the criteria of inclusion and add, if applicable,  the number of registration on an international prospective register of systematic reviews.

Author Response

We are grateful for his comments and suggested implementation. We included details as indicated about diet, SCFAs and exercise.

We also added some explanations about criteria of inclusion; unfortunately, we have not the number of registration on international register.

Reviewer 2 Report

I sugest improve comparation and differences between healthy and not healthy adults

Please, could just reference as short as posible something abou difference between ages, anciant o yapung adults

Althougt could be reference between different countries

You can search reference at

  • Article
  • Published: 22 October 2018

Recovery of gut microbiota of healthy adults following antibiotic exposure

  1. Lynch, S. V. & Pedersen, O. The human intestinal microbiome in health and disease. N. Engl. J. Med. 375, 2369–2379 (2016).
  2. CAS  Google Scholar 
  3. 2.

    Manges, A. R. et al. Comparative metagenomic study of alterations to the intestinal microbiota and risk of nosocomial Clostridum difficile-associated disease. J. Infect. Dis. 202, 1877–1884 (2010).

    Article  Google Scholar 

  4. 3.

    Ley, R. E. et al. Obesity alters gut microbial ecology. Proc. Natl Acad. Sci. USA 102, 11070–11075 (2005).

    CAS  Article  Google Scholar 

  5. 4.

    Turnbaugh, P. J. et al. An obesity-associated gut microbiome with increased capacity for energy harvest. Nature 444, 1027–1031 (2006).

    Article  Google Scholar 

  6. 5.

    Forslund, K. et al. Disentangling type 2 diabetes and metformin treatment signatures in the human gut microbiota. Nature 528, 262–266 (2015).

    CAS  Article  Google Scholar 

  7. 6.

    Karlsson, F. H. et al. Gut metagenome in European women with normal, impaired and diabetic glucose control. Nature 498, 99–103 (2013).

    CAS  Article  Google Scholar 

  8. 7.

    Qin, J. et al. A metagenome-wide association study of gut microbiota in type 2 diabetes. Nature 490, 55–60 (2012).

    CAS  Article  Google Scholar 

  9. 8.

    Craven, M. et al. Inflammation drives dysbiosis and bacterial invasion in murine models of ileal Crohn’s disease. PLoS ONE 7, e41594 (2012).

Author Response

Thank for the interesting literature cited. We expanded the concept of healthy adults and included some references suggested.